# Pharmacogenomics and Drug-Induced Phenoconversion Informed Medication Safety Review in the Management of Pain Control and Quality of Life: A Case Report

**DOI:** 10.3390/jpm12060974

**Published:** 2022-06-15

**Authors:** Selina Muhn, Nishita Shah Amin, Chandni Bardolia, Nicole Del Toro-Pagán, Katie Pizzolato, David Thacker, Jacques Turgeon, Crystal Tomaino, Veronique Michaud

**Affiliations:** 1Office of Translational Research and Residency Programs, Tabula Rasa HealthCare, Moorestown, NJ 08057, USA; smuhn@trhc.com (S.M.); namin@carekinesis.com (N.S.A.); cbardolia@trhc.com (C.B.); npagan@trhc.com (N.D.T.-P.); kpizzolato@trhc.com (K.P.); 2Precision Pharmacotherapy Research & Development Institute, Tabula Rasa HealthCare, Orlando, FL 32827, USA; dthacker@trhc.com (D.T.); jturgeon@trhc.com (J.T.); 3Faculty of Pharmacy, Université de Montréal, Montreal, QC H3T 1J4, Canada; 4VieCare Beaver, Program of All-Inclusive Care for the Elderly (PACE), Lutheran Senior Life, Aliquippa, PA 15001, USA; tomainoc@pmcanetwork.com; 5Research Center of Centre Hospitalier de l’Université de Montréal (CRCHUM), Montreal, QC H2X 0A9, Canada

**Keywords:** pharmacogenomics, pharmacist, opioid, tramadol, amitriptyline, drug–drug–gene interaction, phenoconversion, clinical decision support system

## Abstract

Utilizing pharmacogenomics (PGx) and integrating drug-induced phenoconversion to guide opioid therapies could improve the treatment response and decrease the occurrence of adverse drug events. Genetics contribute to the interindividual differences in opioid response. The purpose of this case report highlights the impact of a PGx-informed medication safety review, assisted by a clinical decision support system, in mitigating the drug–gene and drug–drug–gene interactions (DGI and DDGI, respectively) that increase the risk of an inadequate drug response and adverse drug events (ADEs). This case describes a 69-year-old female who was referred for PGx testing for uncontrolled chronic pain caused by osteoarthritis and neuropathy. The clinical pharmacist reviewed the PGx test results and medication regimen and identified several (DGIs and DDGIs, respectively) at Cytochrome P450 (CYP) 2C19 and CYP2D6. The recommendations were to: (1) switch tramadol to buprenorphine transdermal patch, an opioid with lower potential for ADEs, to mitigate a CYP2D6 DDGI; (2) gradually discontinue amitriptyline to alleviate the risk of anticholinergic side effects, ADEs, and multiple DDGIs; and (3) optimize the pregabalin. The provider and the patient agreed to implement these recommendations. Upon follow-up one month later, the patient reported an improved quality of life and pain control. Following the amitriptyline taper, the patient experienced tremors in the upper and lower extremities. When the perpetrator drug, omeprazole, was stopped, the metabolic capacity was no longer impeded; the patient experienced possible amitriptyline withdrawal symptoms due to the rapid withdrawal of amitriptyline, which was reinitiated and tapered off more slowly. This case report demonstrates a successful PGx-informed medication safety review that considered drug-induced phenoconversion and mitigated the risks of pharmacotherapy failure, ADEs, and opioid misuse.



## 1. Introduction

Chronic pain is a debilitating condition that is prevalent in older adults [1]. Pain management strategies generally begin with non-opioid medications, such as the non-steroidal anti-inflammatory drugs (NSAIDs), then progress to opioids for refractory pain [2]. Adjuvant therapies, such as anticonvulsants or tricyclic antidepressants (TCAs), are often used to specifically manage neuropathic pain [2]. The gene polymorphisms that encode for CYP isoenzymes can alter the drug plasma concentrations, safety, and efficacy of certain analgesic medications (e.g., NSAIDs, opioids, TCAs) [2]. For example, *CYP2D6* is a highly polymorphic gene that codes for CYP2D6, a major enzyme that metabolizes several opioids (e.g., tramadol, codeine, hydrocodone, oxycodone) [3]. The decision to initiate opioids in older adults requires careful consideration due to the increased risk of side effects (e.g., sedation, constipation, physical dependence), as well as changes in the renal and hepatic function that affect the dosing for many of the opioids [1]. The PGx results can help pharmacists personalize the opioid therapies, to achieve optimal efficacy and/or minimize adverse drug events (ADEs) [2].

Polypharmacy may complicate the interpretation of the PGx results, especially when co-administered medications share the same metabolic pathway and affect the drug-induced phenoconversion [4]. Phenoconversion occurs when nongenetic factors, such as concomitant medications, age, and comorbidities, alter the genotype-predicted phenotype of drug-metabolizing enzymes [4]. Assessing for the drug-induced phenoconversion, as opposed to solely relying on genetic results, allows for a more accurate prediction of medication response and an optimization of patient outcomes [4]. The clinical decision support systems (CDSSs) can help to identify drug–drug interactions (DDIs), DGIs, DDGIs, and drug-induced phenoconversion [5]. The CDSS generates a Medication Risk Score (MRS) based on several factors, including the drug interactions [6]. The MRS is associated with health outcomes, including the overall risk of ADEs, falls, and hospitalization [7,8]. The objective is to present a case report with a complex polypharmacy drug regimen supporting the value of a pharmacist-led medication safety review, assisted by a CDSS that incorporates the PGx results and drug-induced phenoconversion.

## 2. Case Presentation

A 69-year-old female patient presented to her healthcare provider with a chief complaint of persistent, severe pain (average = 8, worst = 10), based on the numeric rating scale (NRS) and with an MRS classified as “very high.” The clinical pharmacist reviewed the patient’s medical history and assessed the appropriateness of her current medication regimen, which included tramadol for osteoarthritis and amitriptyline for neuropathic pain (Table 1). The patient scored a 12 on the health-related quality of life questionnaire (EuroQOL-5D; range: 5–15), which assesses health in five dimensions, revealing moderate mobility and self-care difficulties in addition to her extreme pain. A PGx test was proposed to help optimize the patient’s medication therapy. A DNA sample was collected via a buccal swab and was analyzed by a Clinical Laboratory Improvement Amendments-certified laboratory (OneOme, Minneapolis, MN, USA).

Upon review of the PGx results (Table 2) and the medication regimen (Table 3), the clinical pharmacist, aided by a CDSS (MedWise^®^), identified multiple DDGIs affecting the tramadol and perpetrated by the carvedilol, duloxetine, and hydroxyzine. Although the patient was a CYP2D6 normal metabolizer (NM), the plasma concentration of the tramadol’s active metabolite was likely to be lower than expected from the genetic results alone, due to the DDGIs. The drug-induced phenoconversion at CYP2D6 caused her to metabolize the tramadol as an intermediate metabolizer (IM), thereby decreasing the drug’s analgesic effects.

The clinical pharmacist identified the DDGIs affecting the amitriptyline, perpetrated by omeprazole at CYP2C19 and carvedilol, duloxetine, and hydroxyzine at CYP2D6 (Table 3). Although the patient was genetically a *CYP2C19* IM and *CYP2D6* NM, her plasma concentrations of amitriptyline and its active metabolite nortriptyline were likely to be higher and lower, respectively, than predicted from the genetic results alone, due to the DDGIs. The drug-induced phenoconversion at CYP2C19 (to poor metabolizer (PM)) and CYP2D6 (to IM) increased the likelihood of pharmacotherapy failure and ADEs (e.g., cardiac arrythmias) for the amitriptyline. The clinical pharmacist performed a complete medication safety review; however, only recommendations regarding pain, the patient’s primary complaint, will be discussed in detail within this case report. Recommendations addressing the other clinical conditions can be reviewed in Table 4.

During a telephonic consultation, the provider accepted the clinical pharmacist’s recommendations to discontinue the tramadol and initiate the non-CYP2D6 opioid transdermal, buprenorphine. The clinical pharmacist also recommended to gradually taper off and discontinue the amitriptyline, as the risk of continued use outweighed the benefits. Therefore, the provider decided to gradually decrease the dose of amitriptyline over a two-week period until discontinuation (Table 4). The pregabalin and duloxetine were continued to manage the neuropathy pain, and the gabapentin was discontinued to streamline therapy. In addition to these changes, the provider accepted the alternate recommendation to change omeprazole to pantoprazole to mitigate the non-competitive inhibition affecting amitriptyline at CYP2C19 (Table 3).

One month after implementation of the clinical pharmacist’s recommendations, the patient’s pain scores improved by 1.5 points (average = 6.5, worst = 8.5). Her score on the EuroQOL-5D also improved by two points, which indicated improved pain and an ability to perform self-care activities. Her MRS decreased from “very high” to “high,” which was attributed to the mitigated DDGIs, as well as the discontinuation of anticholinergic and sedative medications (amitriptyline, tramadol, gabapentin). Approximately five weeks after the initiation of the amitriptyline taper (Table 4), the patient reported tremors in the arms and legs. The amitriptyline was re-started at 10 mg, and the patient’s symptoms improved.

## 3. Discussion

Despite the significant advances in clinical implementation of PGx and the studies demonstrating inter-individual differences in the safety and efficacy for analgesic medications, a trial-and-error approach when initiating medications is still commonly used by providers [2]. In addition to the genetic variations, DDIs further complicate the pain management in older adults with polypharmacy [2]. Opioids are commonly used to manage osteoarthritic pain that is unsuccessfully managed with first-line analgesics (e.g., NSAIDs) [2].

Tramadol, a prodrug, is a weak µ-opioid agonist that inhibits the reuptake of norepinephrine and serotonin [9]. Tramadol is generally considered an acceptable treatment option for osteoarthritis in certain circumstances, including when NSAIDs are contraindicated or when the patients have an inadequate response to first-line therapies [10]. An NSAID was not initiated due to the patient’s renal and cardiovascular comorbidities (Table 1). However, tramadol should be avoided in older adults due to the risks associated with cognitive impairment and gait disturbances [9]. Tramadol may also lower the seizure threshold, especially when taken in combination with a TCA, such as amitriptyline [9]. Therefore, continued use of tramadol in our patient, who has a history of epilepsy, could increase her risk for seizures. Based on patient-specific factors alone (i.e., age and comorbidities), tramadol is not appropriate for this patient.

Tramadol is metabolized by the CYP2D6 enzyme to an active metabolite, O-desmethyltramadol [3]. The CDSS identified carvedilol, duloxetine, and hydroxyzine as medications with a stronger affinity for the CYP2D6 enzyme than tramadol (Table 3). These interactions resulted in our patient’s CYP2D6 phenotype to be phenoconverted from a NM to an IM for tramadol. This DDGI causes the plasma concentration of tramadol’s active metabolite to be lower than expected from the genetic results alone, which may explain the patient’s uncontrolled pain [11]. According to the Clinical Pharmacogenetics Implementation Consortium (CPIC) Guidelines, tramadol may be used at the recommended age, or weight, specific dosing in CYP2D6 IMs [3]. However, if the patient’s response is inadequate, CPIC recommends a non-codeine opioid [3]. Given that the patient had used other opioids (e.g., hydrocodone/acetaminophen) in the past with minimal success, the provider accepted the recommendation to discontinue the tramadol and initiate transdermal buprenorphine.

Buprenorphine is a schedule III synthetic opioid, with a low potential for physical or psychological dependence, that is used to treat either pain and/or opioid use disorder [12]. Buprenorphine can be safely administered at standard doses in older adults [12]. Furthermore, transdermal buprenorphine is associated with fewer ADEs due, in part, to a ceiling effect that protects against respiratory depression [12]. Buprenorphine is still associated with sedation; older adults should be carefully monitored, especially if they are concomitantly prescribed a benzodiazepine, as in this case [12]. Overall, transdermal buprenorphine may be a relatively safe and effective option for treating chronic pain in older adults.

In addition to replacing tramadol with an alternate opioid, the clinical pharmacist recommended gradually tapering off the amitriptyline. The patient in this case was genetically a *CYP2C19* IM. However, because she was concomitantly taking omeprazole, a non-competitive inhibitor of CYP2C19, the CDSS identified a drug-induced phenoconversion to a CYP2C19 PM for the amitriptyline (Table 3). When this DDGI occurs, we expect that the plasma concentration of the amitriptyline is higher, and the concentration of its active metabolite (nortriptyline) is lower than predicted from the genetic results alone [13]. As a result, she is more likely to experience pharmacotherapy failure and/or ADEs (e.g., blurred vision, dizziness, cardiac arrythmias) [14]. Additionally, the carvedilol, duloxetine, and hydroxyzine have stronger affinities for CYP2D6 than the amitriptyline (Table 3), causing a drug-induced phenoconversion from a CYP2D6 NM to an IM for the amitriptyline. According to the CPIC Guidelines, and considering the phenoconversion of CYP2C19 and CYP2D6, amitriptyline should be avoided in this case [14].

Amitriptyline is a first-line medication for neuropathic pain [9]. However, due to the increased risk of anticholinergic side effects and ADEs (e.g., cognitive impairment and gait disturbances), the American Geriatric Society (AGS) recommends avoiding tertiary amines for patients older than 60 years [9]. Pregabalin and duloxetine, both of which were taken by this patient, are approved for neuropathic pain [9]. Since pain was the primary complaint, her dose of pregabalin was increased to further optimize the therapy for neuropathy. The CPIC recommendation, the AGS Guidelines, the increased risk of ADEs, and the presence of duloxetine and pregabalin, all provided support for the decision to deprescribe amitriptyline for this patient, as the risks outweighed the benefits. Discontinuing the amitriptyline also lowered her MRS, demonstrating a reduced likelihood of ADEs, a lower anticholinergic and sedative burden, and mitigated the DDGIs at CYP2C19 and CYP2D6 [7].

Some individuals experience discontinuation symptoms (e.g., flu-like symptoms, imbalance, nausea, tremors) within seven days of stopping an antidepressant; the onset of symptoms after one week is unusual [15]. To reduce the risk of withdrawal, gradual discontinuation is recommended [15]. The duration of the tapering period may vary, depending on the drug’s half-life [15]. Of note, when the amitriptyline taper was initiated, the omeprazole was also switched to pantoprazole, which is not a mechanism-based inhibitor of CYP2C19 [16]. The offset of the irreversible inhibition depends on the formation rate of a new CYP450 enzyme [17]. A CYP450 enzyme’s half-life is typically 36 h, so it may take three to five days for the enzyme function to return to baseline (for our patient, CYP2C19 IM) following the discontinuation of omeprazole [17]. Based on the PGx results and the assessment of the drug-induced phenoconversion, the health care team concluded that the simultaneous discontinuation of the omeprazole and the gradual tapering of the amitriptyline likely caused an accelerated decrease in the plasma concentration of the amitriptyline, which led to the patient experiencing antidepressant-related withdrawal symptoms. Therefore, a longer taper of the amitriptyline was deemed necessary for this patient, and the amitriptyline 10 mg was re-initiated.

In this case, it was crucial that the healthcare team could easily access and assess both the PGx results and the relevant drug-induced phenoconversion data. Ultimately, the patient’s perception of their quality-of-life and pain (i.e., NRS, EuroQOL-5D) were significantly improved.

## 4. Conclusions

Predicting the safety and efficacy of analgesic medications is complicated by the setting of polypharmacy, drug-induced phenoconversion, and interindividual differences in medication response. This patient case demonstrates the success of a PGx-informed medication safety review, assisted by the CDSS, while accounting for drug interactions and other patient-specific factors (i.e., age, comorbidities). Incorporating sophisticated science and interpretation tools into the medication safety review process can facilitate the individualization of therapy and, such as, in this case, improve the patient’s pain, safety, and quality of life. When implemented, the PGx-informed recommendations made by a clinical pharmacist can optimize medication therapy, ensure efficacy, and reduce the risk of medication-related problems and ADEs. The PGx-informed medication safety reviews that lead to the deprescribing of high-risk medications and the initiation of safer alternatives have the potential to significantly improve care for people with complex drug regimens.

## Figures and Tables

**Table 1 jpm-12-00974-t001:** Patient’s medication list at the time of PGx testing.

Condition	Medication	Dose	Directions
Anxiety	Hydroxyzine	50 mg	1 tablet at bedtime
	Alprazolam	0.5 mg	1 tablet as needed
Atrial fibrillation	Diltiazem	120 mg	1 tablet in the morning
Circulation	Aspirin	81 mg	1 tablet in the morning
	Apixaban	5 mg	1 tablet in the morning and evening
COPD	Albuterol	90 mcg	2 puffs every 6 h as needed
	Ipratropium/albuterol	0.5 mg–3 mg	1 vial via nebulizer four times daily
	Tiotropium	1.25 mcg	2 puffs once daily
Epilepsy	Phenytoin	100 mg	1 tablet in the morning and bedtime
GERD	Omeprazole	40 mg	1 tablet in the morning
Hyperlipidemia	Atorvastatin	40 mg	1 tablet in the morning
Hypertension	Carvedilol	6.25 mg	1 tablet in the morning and bedtime
Hypokalemia	Potassium chloride	20 mEq	1 tablet in the morning
Hypothyroidism	Levothyroxine	25 mcg	1 tablet in the morning
Ischemic cardiomyopathy	Furosemide	80 mg	1 tablet once daily
	Nitroglycerin	0.4 mg	1 tablet every 5 min as needed
	Sotalol	120 mg	1 tablet in the morning and evening
Neuropathy	Gabapentin	100 mg	1 capsule in the morning, evening and bedtime
	Pregabalin	150 mg	1 capsule in the morning and evening
	Duloxetine	60 mg	1 capsule in the morning
	Amitriptyline	75 mg	1 tablet at bedtime
Nutrient deficiency	Multivitamin	N/A	1 tablet in the morning
Osteoarthritis	Tramadol	50 mg	1 tablet in the morning, evening, and bedtime
Osteoporosis	Alendronate	70 mg	1 tablet once a week

Abbreviations: COPD = Chronic obstructive pulmonary disease; GERD = Gastroesophageal reflux disease; PGx = Pharmacogenomics.

**Table 2 jpm-12-00974-t002:** PGx results.

Gene	Genotype	Phenotype Summary
*CYP2C19*	**1|*2*	Intermediate Metabolizer
*CYP2D6*	**2A|*9*	Normal Metabolizer
*CYP2B6*	**1|*5*	Normal Metabolizer
*CYP2C9*	**1|*1*	Normal Metabolizer
*SLCO1B1*	**1B|*1B*	Normal Function

Abbreviations: CYP = Cytochrome P450; SLCO = Solute carrier organic anion transporter; PGx = Pharmacogenomics.

**Table 3 jpm-12-00974-t003:** Summary of affinity and CYP450 metabolic pathway.

Substance	CYP1A2	CYP2B6	CYP2C9 NM → pRM		CYP2C19 IM → pPM		CYP2D6 NM → pIM	CYP3A4	
**Alprazolam**									
**Amitriptyline**									
**Apixaban**									
**Atorvastatin**									
**Carvedilol**									
**Diltiazem**									
**Duloxetine**									
**Hydroxyzine**									
**Omeprazole**									
**Phenytoin**									
**Tramadol**							*		
**Affinity Strengths**	**Weak Substrate**		**Medium Substrate**		**Strong Substrate**		**Inhibitor**	**Inducer**	

Abbreviations: Only CYP-metabolized oral medications are displayed; Derived phenotype → Phenoconverted phenotype; CYP = Cytochrome P450; NM = Normal Metabolizer; RM = Rapid Metabolizer; IM = Intermediate Metabolizer; PM = Poor Metabolizer; p = phenoconversion; ***** = Prodrug.

**Table 4 jpm-12-00974-t004:** Pharmacist’s recommendations and implementations during medication safety review.

Medication	Pharmacist’s Recommendation	Implementation
**Tramadol 50 mg**	Discontinue tramadol and utilize a non-CYP2D6 opioid	Tramadol discontinued and buprenorphine transdermal patch 5 mcg/h weekly initiated
**Amitriptyline 75 mg**	Taper off amitriptyline to mitigate risk of ADEs and pharmacotherapy failure	Amitriptyline 50 mg for 1 week, 25 mg for 1 week, then discontinued
**Omeprazole 40 mg**	Switch to pantoprazole 40 mg to mitigate non-competitive inhibition at CYP2C19	Switched to pantoprazole 40 mg
**Amitriptyline 75 mg, Furosemide 80 mg, Hydroxyzine 50 mg, Omeprazole 40 mg, Sotalol 120 mg, Tramadol 50 mg**	Re-evaluate the need for QT-prolonging medications and obtain ECG	Will monitor ECG
**Atorvastatin 40 mg**	Switch to pravastatin to mitigate drug interaction with phenytoin	Switched to pravastatin 40 mg
**Pregabalin 150 mg and Gabapentin 100 mg**	Utilize either pregabalin or gabapentin to mitigate sedative burden	Gabapentin discontinued and pregabalin dose increased from 150 mg to 225 mg
**Alprazolam 0.5 mg**	Switch to lorazepam to mitigate sedative burden and drug interaction at CYP3A4	Patient declined
**Hydroxyzine 50 mg**	Taper off hydroxyzine to mitigate anticholinergic and sedative burden	Patient declined
**Diltiazem 120 mg, Carvedilol 6.25 mg, Sotalol 120 mg**	Consult cardiology to evaluate appropriateness of cardiovascular drug regimen	Cardiology consulted

Abbreviations: ADEs = Adverse drug events; CYP = Cytochrome P450; ECG = Electrocardiogram.

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
