# Peer review of "Pharmacogenomics and Drug-Induced Phenoconversion Informed Medication Safety Review in the Management of Pain Control and Quality of Life: A Case Report"

_jpm, 2022, doi:10.3390/jpm12060974_

Round 1
Reviewer 1 Report
In this case report, several drug-gene and drug-drug-gene interactions were identified by the clinical pharmacist after reviewing the pharmacogenomics test results and medication regimen in a 69-year-old female who suffered from uncontrolled chronic pain. Incorporating knowledge of pharmacogenomics results and drug-induced phenoconversion into the medication safety review process significantly improved patient pain, safety, and quality of life. In my opinion, it is worth to be published.
Author Response
Reviewer #1
- In this case report, several drug-gene and drug-drug-gene interactions were identified by the clinical pharmacist after reviewing the pharmacogenomics test results and medication regimen in a 69-year-old female who suffered from uncontrolled chronic pain. Incorporating knowledge of pharmacogenomics results and drug-induced phenoconversion into the medication safety review process significantly improved patient pain, safety, and quality of life. In my opinion, it is worth to be published.
We would like to thank the reviewer for taking time to revise our manuscript and his/her positive feedback.
Reviewer 2 Report
With real interest, I read a case report jpm-1758368. It is an article very well matching the scope of the journal, at last in my opinion.
There is not much to correct/comment on here. Thus, I have minor comments only:
11. The title reads more like a title of a review, not a case report. Please, consider changing it.
22. Lines 16-20 “The purpose of this case report is to demonstrate the impact of a PGx-informed medication safety review, assisted by a clinical decision support system, in mitigating drug-gene and drug-drug-gene interactions (DGI and DDGI, respectively) that increase the risk of inadequate drug response and adverse drug events (ADEs).” and lines 62-64 “The objective of this case report is to demonstrate the value of a pharmacist-led medication safety review, assisted by a CDSS that incorporates PGx results and drug-induced phenoconversion.”. The purpose/objective of this case report is to present a case. The rest are conclusions based on the case presentation. Please, consider rewriting.
33. The names of the genes should be written in italics also in the main text, e.g. “CYP2D6” in line 45. And throughout the text, if any similar cases.
Author Response
Reviewer #2
With real interest, I read a case report jpm-1758368. It is an article very well matching the scope of the journal, at least in my opinion. There is not much to correct/comment on here. Thus, I have minor comments only:
We would like to thank the reviewer for his/her review and comments pertaining to this manuscript.
- The title reads more like a title of a review, not a case report. Please, consider changing it.
Title was changed to the following: “Pharmacogenomics and drug-induced phenoconversion informed medication safety review in the management of pain and quality of life: a case report.”
- Lines 16-20 “The purpose of this case report is to demonstrate the impact of a PGx-informed medication safety review, assisted by a clinical decision support system, in mitigating drug-gene and drug-drug-gene interactions (DGI and DDGI, respectively) that increase the risk of inadequate drug response and adverse drug events (ADEs).” and lines 62-64 “The objective of this case report is to demonstrate the value of a pharmacist-led medication safety review, assisted by a CDSS that incorporates PGx results and drug-induced phenoconversion.”. The purpose/objective of this case report is to present a case. The rest are conclusions based on the case presentation. Please, consider rewriting.
The sentence (previous lines 16-20) was changed to the following: “This case report highlights the impact of a PGx-informed medication safety review, assisted by a clinical decision support system (CDSS), in mitigating DGIs and DDGIs that increase the risk of inadequate drug response and ADEs.”
The second sentence was rewritten as requested (previous lines 62-64): “The objective is to present a case report with a complex polypharmacy drug regimen supporting the value of a pharmacist-led medication safety review, assisted by a CDSS that incorporates PGx results and drug-induced phenoconversion.”
- The names of the genes should be written in italics also in the main text, e.g. “CYP2D6” in line 45. And throughout the text, if any similar cases.
Names of genes were italicized throughout the text.